# Stress Response to Climate Change and Postharvest Handling in Two Differently Pigmented Lettuce Genotypes: Impact on *Alternaria alternata* Invasion and Mycotoxin Production

**DOI:** 10.3390/plants12061304

**Published:** 2023-03-14

**Authors:** Jon Miranda-Apodaca, Unai Artetxe, Iratxe Aguado, Leire Martin-Souto, Andoni Ramirez-Garcia, Maite Lacuesta, José María Becerril, Andone Estonba, Amaia Ortiz-Barredo, Antonio Hernández, Iratxe Zarraonaindia, Usue Pérez-López

**Affiliations:** 1Department of Plant Biology and Ecology, Faculty of Science and Technology, University of the Basque Country (UPV/EHU), Barrio Sarriena s/n, 48940 Leioa, Spain; 2Applied Genomics and Bioinformatics Research Group, Department of Genetics, Physical Anthropology and Animal Physiology, University of the Basque Country (UPV/EHU), Barrio Sarriena s/n, 48940 Leioa, Spain; 3Fungal and Bacterial Biomics Research Group, Department of Immunology, Microbiology and Parasitology, Faculty of Science and Technology, University of the Basque Country (UPV/EHU), Barrio Sarriena s/n, 48940 Leioa, Spain; 4Department of Plant Biology and Ecology, Faculty of Pharmacy, University of the Basque Country (UPV/EHU), 01006 Vitoria-Gasteiz, Spain; 5NEIKER-Basque Institute for Agricultural Research and Development, 01080 Vitoria-Gasteiz, Spain; 6IKERBASQUE, Basque Foundation for Science, Plaza Euskadi 5, 48009 Bilbao, Spain

**Keywords:** *Alternaria alternata*, climate change scenario, defense mechanisms, differently pigmented lettuce, mycotoxins, postharvest handling

## Abstract

Many species of *Alternaria* are important pathogens that cause plant diseases and postharvest rots. They lead to significant economic losses in agriculture and affect human and animal health due to their capacity to produce mycotoxins. Therefore, it is necessary to study the factors that can result in an increase in *A*. *alternata*. In this study, we discuss the mechanism by which phenol content protects from *A*. *alternata*, since the red oak leaf cultivar (containing higher phenols) showed lower invasion than the green one, Batavia, and no mycotoxin production. A climate change scenario enhanced fungal growth in the most susceptible cultivar, green lettuce, likely because elevated temperature and CO_2_ levels decrease plant N content, modifying the C/N ratio. Finally, while the abundance of the fungi was maintained at similar levels after keeping the lettuces for four days at 4 °C, this postharvest handling triggered TeA and TEN mycotoxin synthesis, but only in the green cultivar. Therefore, the results demonstrated that invasion and mycotoxin production are cultivar- and temperature-dependent. Further research should be directed to search for resistant cultivars and effective postharvest strategies to reduce the toxicological risk and economic losses related to this fungus, which are expected to increase in a climate change scenario.

## 1. Introduction

*Alternaria* is a ubiquitous fungal genera, and is able to grow in diverse environments. It is the cause of significant economic losses in agriculture, as many of the species within *Alternaria* are phytopathogens, generating important plant diseases and postharvest rots in produce [1]. In addition, some *Alternaria* spp. can produce a variety of secondary metabolites (more than 70) with toxic properties for plants and/or animals, which are known as mycotoxins [2,3]. Some of them are of clinical significance as they can cause respiratory tract infections and asthma in humans and animals, and even cancer [4]. The most relevant mycotoxins causing health problems are alternariol (AOH), alternariol monomethyl ether (AME), tentoxin (TEN), tenuazonic acid (TeA), altenuene (ALT), altertoxin I and II, stemphyltoxin III, and *Alternaria alternata* f. sp. *lycopersici* toxins (AAL toxins) [5,6,7]. Within the genera, *A. alternata* is considered to be the most important mycotoxin-producing species [8]. Importantly, 400 species of plants have been reported to be susceptible to infection by *Alternaria*, of which *A. alternata* is capable of infecting a quarter [9], demonstrating that it is a highly successful pathogen with a wide host range. The internal transcribed spacer (ITS) gene is one of the most widely targeted molecular markers for its species identification; however, it does not allow the distinction of all members within section *Alternaria*. Therefore, in such cases, a multi-locus approach or a next generation sequencing strategy have been recommended [10,11].

Due to all of the above, in recent years the research community has been making a significant effort to study the occurrence of these fungi in different plant species, as well as their toxin production capacity [12,13,14,15,16,17,18,19] and their presence in the markets [20,21,22,23]. Furthermore, the European Food Safety Authority (EFSA) performed a dietary exposure assessment to *Alternaria* toxins in the European population [4]. These studies highlighted the need for standardizing protocols for sample collection and sample processing to make *Alternaria* abundance and evaluation of its toxins comparable across studies.

Similarly, the agriculture industry is in need of exploring alternatives to overcome the challenges associated with the adverse impact caused by pathogenic fungi. The incidence and prevalence of foodborne diseases are highly linked to environmental conditions, climate change being a current global concern affecting not only crop production, but also food security and safety [24,25]. It is likely that elevated CO_2_ and higher temperatures will increase fungi invasions and toxin production, although the response will be dependent on the host/pathogen combination [26,27,28]. In particular, future climatic conditions are expected to change *Alternaria* incidence and toxin production [15,29,30,31], because elevated CO_2_ alters phytohormone and reactive oxygen signaling, secondary metabolism, and defense-associated development such as stomatal responses in the host [32], resulting in a higher vulnerability of hosts to phytopathogens such as *Alternaria*. Possible strategies to limit *Alternaria* infection include: (1) obtaining resistant cultivars in breeding programs. In this regard, the studies conducted so far point to the plant’s ability to increase phenolic compounds and its antioxidant metabolism as good traits to select for [33,34,35]; (2) selecting adequate cultivation and field management techniques in order to reduce fungi invasion [14,36]. Cultivation practices, such as maintaining a healthy field and crop vigor and removing infected plant debris, are helpful in controlling fungi invasion, but are not always sufficient; (3) improving postharvest handling, since vegetables can also be contaminated by *A. alternata* mycotoxins during this period [37]. Postharvest handling aims to preserve the freshness and quality of agricultural products, and includes processes to minimize the growth of fungus and/or mycotoxin production. The postharvest fungal growth and mycotoxin production mainly depends on temperature management, water activity, and technological treatments (chemicals, edible coatings, and gaseous treatments) to prolong postharvest quality [38]. High temperatures during storage are known to stimulate *A. alternata* growth in several species [39,40], but it is not clear whether it increases mycotoxin production in the same way. While cold storage is a common strategy to reduce fungal growth and mycotoxin production [38], several mycotoxins have shown to be more abundant at lower temperatures [40,41]. However, the majority of conclusions are based on studies conducted either in tomato or potato plants, while other crops/vegetables have received less attention.

Lettuce (*Lactuca sativa* L.), is a major food crop within the European Union, according to European Commission Eurostat (2023). Its production in the European Union reached around two and a half million tons in 2021, with Spain being the main producer (1,063,000 tons) and exporter. Its consumption has increased considerably due to its high nutritional value; it contains high amounts of phytochemicals, including vitamins, carotenoids, and other antioxidants [42,43]. *Alternaria* has previously been recorded occasionally in lettuce crops, and this species is frequently chosen as a model to assess the potential impact of increased CO_2_ and temperature on disease incidence and severity under a controlled environment [44].

In the present study, two lettuce cultivars were subjected to *A. alternata* inoculation and grown in different environmental conditions, simulating either the present atmospheric temperature and CO_2_ conditions or a climate change scenario. The two lettuce cultivars chosen for the study contain different phenol concentrations [45]; therefore, we hypothesize that they will cope differently with the fungal infection, as previous works have stated that phenol concentration is a resistant trait against phytopathogens in tomato [33,34]. The main objectives of this work are: (1) to explore lettuce defense mechanisms and traits that can make cultivars more resistant to *A. alternata* infection; (2) to address how a future climate change scenario would affect the growth of this fungus and its toxin production, and whether it would have a different impact depending on the host cultivar; (3) to study the potentiality of low-temperature storage as a postharvest handling strategy to reduce *A. alternata* growth and toxin production.

## 2. Results

### 2.1. Response of Fresh Lettuce to A. alternata Infection: Effect of Host Cultivar and Environmental Conditions on Fungi Growth, Mycotoxin Production, and Antioxidant Enzyme Activity

The obtained fresh biomass of the green cultivar was 80 g FW plant^−1^, and it was not affected either by *A*. *alternata* inoculum or by environmental conditions. The red cultivar followed a similar trend, although its biomass in all studied conditions was higher (Figure 1A). Regarding water content, it was around 93% in all treatments, a value considered adequately fresh (Figure 1B).

The presence of the species *A. alternata* was negligible in non-inoculated samples regardless of lettuce cultivar or environmental condition, confirming that lettuces were not originally contaminated by native *A. alternata* and that there was no cross-contamination between inoculated and non-inoculated samples. However, the fungus reached a maximum of 2452 genome copies in inoculated samples (Table 1). The color of the lettuce had a strong influence on the growth of the fungus, with the Batavia cultivar (green) showing significantly higher amounts than the red (Table 1; Mann–Whitney *p* < 0.001). Importantly, the red cultivar, despite having been inoculated, was able to maintain *A. alternata* in low abundances (genome copies in ATC = 16.4 ± 6.8 and in ETC = 0.7 ± 0.5, Table 1). No significant differences in the fungal growth were found between inoculated samples subjected to ATC or ETC conditions (*p* = 0.645). In addition, within the green cultivar, ETC samples exhibited higher mean abundances (ETC = 671.7 ± 565.2 and, ATC = 56.6 ± 26.5, Table 1) but the comparison was not significant (*p* = 0.200) due to the high standard error within each of the conditions.

Regarding mycotoxins, none of the fresh lettuces, despite the different environmental conditions or cultivar color, showed AME, AOH, TeA, or TEN mycotoxins (Table 1).

However, the antioxidant system was modified by the studied factors. SOD activity was affected by lettuce color (*p* < 0.05) and by *A. alternata* inoculum (*p* < 0.05), but not by environmental conditions (*p* = 0.737). However, an interaction between environmental conditions and color was detected (*p* < 0.05). In fact, SOD activity was generally higher in the green cultivar than in the red one (Figure 2A). Regarding the inoculum treatment, under ATC conditions, the lettuces inoculated with *A*. *alternata* increased SOD activity by 19% and 17% in the green- and red-pigmented cultivars, respectively, compared to the plants that were not inoculated. When the lettuces were grown under ETC conditions, in the green cultivar, independently to the inoculation treatment (inoculated or non-inoculated), the SOD activity was similar to ATC non-inoculated green lettuces. However, in the red cultivar, under ETC conditions, we detected 25% increases in both inoculated and in non-inoculated leaves compared to the ATC non-inoculated red cultivar.

APX activity was affected by lettuce color (*p* < 0.05) and by environmental conditions (*p* < 0.05), but not by *A. alternata* inoculum (*p* = 0.931). We also detected an interaction between environmental conditions and color (*p* < 0.05). As occurred for SOD, in the green cultivar, the activity was higher than in the red cultivar (Figure 2B). Moreover, in the red cultivar, ETC conditions decreased this activity even more, although no differences between inoculated and non-inoculated leaves were detected.

Similarly to APX, MDHAR activity was also at its highest in the green cultivar (*p* < 0.05; Figure 2C), decreasing by 29% under ETC conditions in plants that were not inoculated compared to ATC plants. This pattern was also observed in the red cultivar. In the case of the inoculated plants, no differences were detected, but we observed an interaction between *A. alternata* inoculum and environmental conditions (*p* < 0.05).

DHAR activity was affected by lettuce color (*p* < 0.05) and by *A. alternata* inoculum (*p* < 0.05), but not by environmental conditions (*p* = 0.674). In the green cultivar, the activity was higher than in the red cultivar (Figure 2D). In addition, in the green cultivar, we detected increases in the activity when inoculated plants were grown under ETC conditions. Although, in the red cultivar, no changes were detected, with all of the treatments, the inoculum treatment tended to increase the activity, albeit not significantly.

Regarding reduced ascorbate, no significant differences were detected between cultivars (*p* = 0.064), and it was not affected by either the *A*. *alternata* inoculum (*p* = 0.703) or by the environmental conditions (*p* = 0.637; Figure 3A).

The analysis of total phenol concentration indicated that they were affected by lettuce color (*p* < 0.05), being higher in the red cultivar than in the green cultivar (Figure 3B). We also detected an interaction between *A. alternata* inoculum and color (*p* < 0.05) and a triple interaction between *A. alternata* inoculum, color, and environmental conditions (*p* < 0.05). In fact, in the green cultivar, under ATC conditions, the inoculum increased the phenol concentration by 158% under ATC conditions and by 32% under ETC conditions. In the red cultivar, phenol concentration was not affected either by inoculum treatment or by environmental conditions (*p* > 0.05).

### 2.2. Effect of Cold Storage on the Growth of Fungi and Mycotoxins in Lettuce Plants Grown under Different Environmental Conditions

Half of the lettuces of the experiment were wrapped in plastic bags and maintained at 4 °C for four days, simulating cold storage handling, with the aim of studying how low temperature would affect the occurrence of fungi and mycotoxins.

Despite the high variability in *A. alternata* abundance between plant replicates, overall, the mean abundance of the fungi was similar in lettuces stored for four days at 4 °C with respect to the fresh samples (Table 1, Mann–Whitney *p* = 0.724). Within the cold-stored samples, the green cultivar under ETC conditions showed lower, but not significant, mean *A. alternata* abundances than samples grown under ATC conditions (ETC = 17.7 ± 5.4 vs. ATC = 662.8 ± 596.7, *p* = 0.057).

Interestingly, although *A. alternata* was noticed in most cold-stored samples, mycotoxin accumulation, particularly TeA and TEN, was only detected under specific conditions, i.e., cold-stored *A. alternata*-inoculated green lettuces (Table 1). Neither fresh nor cold-stored red lettuce samples showed mycotoxins. This means that in our experimental design, with a confluence of only three factors (green variety, inoculation, and cold storage at 4 °C for four days), *A. alternata* was able to induce mycotoxin production at concentrations < 1 mg kg^−1^ in lettuces (Table 1). Even so, only half of the replicates (2 out of 4) in ETC, and 1 out 4 in ATC conditions showed mycotoxins, the concentration being higher (but not statistically significant) in ETC conditions as compared to ATC. Importantly, while cold storage favored TeA and TEN synthesis, AME and AOH were not produced.

## 3. Discussion

### 3.1. Lettuce Cultivar Color, Rather Than Environmental Conditions, Determines A. alternata Invasion

In this study, two lettuce cultivars, green and red, were subjected to *A. alternata* inoculation and different environmental conditions, simulating the present and a potential climate change scenario. The green lettuce cultivar increased total phenols in response to fungal infection, but the red lettuce did not. Thus, although the increase rate was higher in the green cultivar, a higher rate of *A. alternata* invasion was reported on that cultivar. In comparison, it contained lower total phenols than the red lettuce did.

It is known that plants are able to defend themselves from pathogenic fungi by using their basal defense mechanisms or activating other specific ones [46]. Awan et al. [34] stated that phenolic compounds in infected leaves could be used to assist with screening of pathogen-resistant genotypes, supporting the idea that phenols function as defensive mechanisms. Similarly, Wianowska et al. [47] proposed that phenolic acids were responsible for the activity of the walnut green husk, an agro-forest waste generated in the walnut (*Juglans regia* L.) harvest that could be valued as a source of natural compounds with antioxidant and antimicrobial properties. These authors demonstrated that extracts of walnut green husk enriched in phenols could inhibit mycelial growth of *A. alternata*, among other pathogens. This effect could be due to the capability of these compounds to inhibit the germination of fungal spores [48]. It has been concretely demonstrated that methyl *p*-coumarate has antifungal activity against *A. alternata*, inhibiting mycelial growth and spore germination [49]. Similarly to those studies’ conclusions, our results support that phenols are key compounds of the plant defense system during *A. alternata* pathogenic attacks.

Besides phenols, additional strategies likely contributed to limiting *A. alternata* invasion. In fact, the green cultivar showed higher absolute activities of the SOD, APX, MDHAR, and DHAR enzymes than the red cultivar. However, these higher basal values of the antioxidant system were not as efficient in controlling the growth of *A. alternata;* as previously mentioned, the green cultivar showed the most fungal growth. Moreover, ascorbate’s role in the defense against *A. alternata* can be discarded at ambient CO_2_ conditions since it remained constant, as did the enzymes in charge of regenerating it, namely, MDHAR and DHAR. However, a lack of coordination between SOD and APX activities was observed in green inoculated lettuces. The observed increase in SOD activity and the maintenance of APX may provoke an increase in H_2_O_2_, as was demonstrated by Meena et al. [9] in tomato infected by *A. alternata*. The extra generated H_2_O_2_ could help lettuce to cope with fungi, since it has been observed that H_2_O_2_ is able to inhibit fungal growth directly, or it may generate other free radicals with antimicrobial activities against fungi [50]. In fact, it has been evidenced that *Vitis vinifera* cultivars that were able to accumulate H_2_O_2_ were more resistant to phytopathogenic fungi than the ones that were not able to accumulate it [51].

Regarding the possible impact of future climate conditions, its effect was more pronounced on the green cultivar, in which a higher *A. alternata* invasion was observed in the climate change scenario (Table 1). Similar results have been detected in *Eruca sativa* (cultivated rocket) for *Alternaria* leaf spots [52]. Siciliano et al. [15] also demonstrated that carbon dioxide and temperature influenced *Alternaria* sp. invasion, observing the highest disease severity at 22–26 °C and 800–850 ppm of CO_2_ [15]. In fact, Canihos et al. [53] reported that the optimum temperature range for *Alternaria* was between 23–27 °C, with a maximum at 27 °C, which is similar to the conditions used to simulate a climate change scenario in this study (average day/night temperature of 26/22 °C and CO_2_ concentration of 700 μmol mol^−1^).

It has frequently been demonstrated that plants growing under elevated CO_2_ reduce their nitrogen content, showing a higher C/N ratio as a result of higher photosynthetic rates and the N dilution effect [54]. It is also known that high N levels prolong plant vigor, reducing their susceptibility to infestation by *Alternaria* [55]. Thus, the greater invasion of fungi under ETC conditions could be due to low N availability. In fact, it has been described that leaf changes associated with elevated CO_2_ increase *A. alternata* spore production [29]. In addition, plant respiration rates usually increase under elevated temperatures [56], reducing the extra carbohydrate availability. Thus, the ETC plants, aside from disposing of less N, would not be able to direct carbohydrates to produce phenols. Our results support this hypothesis, as under ATC conditions, a sharp increase in phenol concentration was observed, while under ETC, the increase was less severe. Furthermore, it has been demonstrated that *A. alternata* that causes early blight is a low-sugar disease [55] and, thus, any factor (e.g., elevated temperature) that decreases the sugar content would result in increased severity of the pathogenicity of *A. alternata*.

While the inoculated fresh lettuces studied in the present work showed a significant higher abundance of *A. alternata*, as shown by qPCR, no TeA, TEN, AOH, or AME mycotoxins were produced either in the present climatic scenario or in elevated CO_2_ and temperature conditions (climate change scenario, Table 1), thus confirming that the presence of *A. alternata* does not indicate mycotoxin production [15].

### 3.2. Storage Temperature, Rather Than Fungal Occurrence, Determines Mycotoxin Production

The concentrations of TeA and TEN mycotoxins detected in the lettuces in this study were in the low range of concentration among those described in the literature [4]; thus, it would be practically impossible to reach the threshold of toxicological concern values for these two toxins (>1500 ng/kg body weight [5]). Mycotoxins were only found in cold-stored samples, but its detection was not associated with the amount of *A. alternaria* present in the lettuce leaves. For instance, TEN and TeA mycotoxins were identified in a sample with as few as 16 genome copies of *A. alternaria* (Table 1), while other samples with higher fungi abundances in the same environmental conditions were mycotoxin-free. Therefore, as Siciliano et al. [15] described before for *Alternaria*, our results confirm that mycotoxin production is not directly related to fungal abundance.

Importantly, the cold storage triggered the synthesis of TEN and TeA mycotoxins. Previous studies have pointed out that 22–30 °C is the optimal growth temperature for *Alternaria* sp., but this fungus is able to grow even at low temperatures during cold postharvest storage [40,57], causing the spoilage of fruits. Although the fungal growth is inhibited at 4 °C, mycotoxins can accumulate rapidly [39]. As is similar to our findings, Oviedo et al. [41] stated that mycotoxin production in soybean is a temperature-dependent process, as TeA was greater at 28 °C, but the synthesis of both AME and AOH was favored at 4 °C. However, some authors have described restricted *Alternaria* mycotoxin production below 6 °C [58,59].

Mycotoxins were only produced in lettuce that was cold-stored for 4 days, grown both in present and in climatic change environmental conditions, but their synthesis was only detected in the green cultivar. This agrees with previous findings by other authors [15,60,61] that mycotoxin production is host-dependent. In this particular study, the green cultivar (characterized by less phenols) was more prone to mycotoxin production. Moreover, the type of mycotoxin that is produced depends on both the *A. alternata* strain and the host species. We detected TEN and TeA, but not AME or AOH, in the green lettuce. On the contrary, Oviedo et al. [41] detected AME and AOH synthesis at 4 °C, but not TEN and TeA in soybean, while Meng et al. [39] found TeA, AOH, and AME in inoculated yellow peaches when stored at 4 and 28 °C.

Environmental conditions could also play a role in the synthesis of toxins, since a higher accumulation of TeA and TEN was measured in plants grown under the climate change scenario. However, it must be noted that this accumulation was not statistically significant. As previously mentioned, plant metabolism is different in high CO_2_ conditions [29,62], affecting the C/N ratio. This ratio is important for the production of secondary metabolites, as N content may condition fungal invasion and the plant’s response. Similarly, the C/N ratio also determines mycotoxin production, with high C/N ratios promoting their synthesis. For instance, Casas López et al. [63] studied the influence of different C/N ratios on lovastatin production by *Aspergillus*, and found that the presence of excess carbon under nitrogen limitation greatly enhanced the production of this mycotoxin, its optimal production being at a C/N ratio ~40. These authors concluded that lovastatin production is associated with the nitrogen-limited stationary growth phase, in which the excess of carbon can be channeled into secondary metabolism. The same can be applied to the mycotoxin production of *A. alternata*, as Brzonkalik et al. [64] clearly showed that a secondary metabolite character of alternariol was produced in the stationary growth phase following N depletion. In our experiment, the potential excess of carbon and decrease in N associated with climate change environmental conditions is likely responsible for limiting the accumulation of phenols, making those plants less resistant to the fungi growth, and at the same time for favoring the production of mycotoxins.

## 4. Materials and Methods

### 4.1. Experimental Design

The experiment was performed with two *Lactuca sativa* L. cultivars (cv. Batavia and cv. oak leaf; [62]). Henceforth, cv. Batavia will be referred to as green lettuce and cv. oak leaf will be referred to as red lettuce. Plants were sown in a mixture of perlite/vermiculite (3:1) in 1.4 L pots in an environmentally controlled growth chamber. They were placed under a daily regime of 14 h of light provided by warm-white fluorescent lamps (Philips TL5 HO 54W 840) and 10 h of darkness, in two conditions: (i) the present climatic scenario, with the current atmospheric temperature and CO_2_ conditions (ATC: day/night temperature of 22/18 °C, day/night relative humidity of 60/80%, and a CO_2_ concentration of 400 μmol mol^−1^); and (ii) the climate change scenario, with elevated temperature and CO_2_ conditions (ETC: day/night temperature of 26/22 °C, day/night relative humidity of 60/80%, and CO_2_ concentration of 700 μmol mol^−1^). Both environmental conditions were controlled in the chamber from sowing until the end of the experiment, for 24 h a day. In both ATC and ETC conditions, the photosynthetically active radiation (PAR) was approximately 400 μmol photons m^−2^ s^−1^ and it was measured with a Li-250A light meter equipped with a quantum sensor (Li-190SA, Li-Cor, Inc. Lincoln, NE, USA).

Seedlings were watered with Hoagland‘s solution [65] every 2 days until 38 days after sowing (DAS) was reached. From this day onward, half of the pots (n = 32) were inoculated with *A*. *alternata* (Aa+), while the other half were not inoculated (Aa−). Inoculated plants were brushed (on both sides of the leaflets) with a spore suspension of the pathogen (1.0 × 10^6^ conidia mL^−1^), and the non-inoculated plants were brushed (on both sides of the leaflets) with sterilized water. The plants were covered with plastic bags for 48 h to maintain suitable humidity levels (65–70%). At this point, there were lettuce plants growing in four conditions: (1) lettuces grown in a present climatic scenario and not inoculated with *A*. *alternata* (ATC and Aa−); (2) lettuces grown in a present climatic scenario and inoculated with *A*. *alternata* (ATC and Aa+); (3) lettuces grown in a climate change scenario and not inoculated with *A*. *alternata* (ETC and Aa−); and (4) lettuces grown in a climate change scenario and inoculated with *A*. *alternata* (ETC and Aa+). Plants were harvested 14 days later (52 DAS). Half of the harvested plants were frozen in liquid nitrogen (hereafter referred to as fresh lettuce) for further measurements of their antioxidant systems, *A. alternata* abundance, and mycotoxin presence. The other half of the lettuces were placed into plastic bags and maintained at 4 °C for four days in order to simulate fridge conditions (hereafter referred to as cold-stored lettuce). After four days, cold-stored leaves were collected, introduced into liquid nitrogen, and stored at −80 °C until they were processed for the purpose of measuring the abundance of *A*. *alternata* and mycotoxins.

### 4.2. Fungal Culture and Inoculum Preparation

To isolate the *A. alternata* strain, air-dried barley grains showing premature ripening and darkening of the ears at harvest were plated on a potato dextrose agar supplemented with antibiotics to inhibit bacterial growth in room conditions (mPDA). The isolates were examined under a stereomicroscope and were assigned to the *Alternaria* genus, taking into account that the key taxonomic feature of the genus *Alternaria* is the production of large, multicellular, dark-colored (melanized) conidia with longitudinal as well as transverse septa (phaeodictyospores) [1]. After single conidial isolates were obtained by streaking a conidial suspension on PDA, the sample was grown on mPDA for molecular identification. The total genomic DNA was extracted from the isolates using an NZY Plant/Fungi gDNA Isolation Kit (NZYTech), and DNA concentration was measured by a NanoDrop^TM^ 1000 Spectrophotometer (Thermo Scientific, Waltham, MA, USA) in order to then amplify the ITS gene and Sanger sequence PCR products. The universal fungal primers ITS1′ (5′-TCCGTAGGTGAACCTGCGG-3) and ITS4′ (5′-TCCTCCGCTTATTGATATGC-3′) were used to amplify the ITS1-5.8S-ITS2 region, following the procedure used by White et al. [66]. In addition, primers AAF2 (5′-TGCAATCAGCGTCAGTAACAAAT-3′) and AAR3 (5′-ATGGATGCTAGACCTTTGCTGAT-3′) [67] were used following the conditions described by Basım et al. [68]. The amplified PCR products were sequenced in an ABI PRISM 3100 (Thermo Scientific, Waltham, MA, USA) by the Sequencing and Genotyping Unit of the University of the Basque Country, (SGIker). DNA sequences were quality checked with Bio-Edit [69], and BLASTn v2.12.0 was used to search for species identity in NCBI GenBank. The isolate was identified as *A. alternata* after similarity analysis (NCBI accession number OQ434290).

A fresh spore suspension was prepared by scraping twice-grown agar plates with sterile distilled water (sDW). The harvested conidia suspension was filtered through a sterile gauze and centrifuged at 4400× *g* and 4 °C for 30 min; then, the concentration was adjusted using a hemocytometer for a final spore density of 1.0 × 10^6^ conidia mL^−1^ in sDW.

### 4.3. Growth and Freshness Analysis

Biomass production and freshness, determined by leaf water content, were measured at 52 DAS in terms of fresh weight (FW), following the protocol outlined by Pérez-López et al. [62].

### 4.4. Antioxidant Metabolism Analysis

The activities of superoxide dismutase (SOD), ascorbate peroxidase (APX), monodehydroascorbate reductase (MDHAR), and dehydroascorbate reductase (DHAR) were essentially determined according to Pérez-López et al. [70]. The protein concentration was measured according to Bradford’s method [71] using bovine serum albumin as a standard. Ascorbate and total phenols were measured according to the process of Pérez-López et al. [72].

### 4.5. Alternaria Alternata Abundance by qPCR

Leaves samples (n = 64) were homogenized with a manual stomacher, and 200 mg were used for total microbial DNA extraction with the Dneasy Plant Mini kit (QIAGEN) following the manufacturer’s instructions, with the addition of a bead-beating lysis on tubes containing 0.1 mg glass beads in a Precellys (6500 rpm 3 × 30 s). DNA was quantified using a NanoDrop Lite Spectrophotometer (Thermo Scientific, Waltham, MA, USA). An AltAlt dtec-qPCR Test kit (Genetic PCR Solutions), specifically designed to identify and quantify *A. alternata* and excluding other species from the genera, was used following the manufacturer’s instructions. For the qPCR, a 250 ng DNA template was used in duplicate. The standard curve was prepared by dilution of the standard template provided in the kit. The number of *A. alternaria* genome copies were calculated using the *Ct* obtained from the qPCR with the following formula:(1)Ct=Yintercept+Slope×log(copynumber)
(2)$$Copynumber=10^{\frac{\left(Ct-Yintercept\right)}{Slope}}$$

The qPCR platform which we used was QuantStudio™ 3 System (Applied Biosystem, Waltham, MA, USA), and the results were analyzed with the QuantStudio™ Design & Analysis Software v1.5.1. The significance values of the *A. alternata* abundance differences between groups were obtained using the Mann–Whitney U test in SPSS v28 (IBM Corp, Armonk, NY, USA).

### 4.6. Extraction and Purification of Mycotoxins

The mycotoxin extraction and quantification method was based on the UNE-EN15662:2019 standard (European Commission, Brussels, Belgium, 2006).

Extraction kits from Agilent Technologies (Santa Clara, CA, USA) were used for extraction with the QuEChERS-dSPE method. First, 10 g of previously frozen lettuce samples were weighed into 50 mL centrifuge tubes and 10 mL of water was added. Afterwards, extraction was carried out with 10 mL of acetonitrile acidified with 2% formic acid (HCOOH). As a result of this acidification, the deprotonation of acid mycotoxins in the extracting organic phase (e.g., TeA and TEN) was avoided and the turbidity provided by soluble proteins was eliminated. Samples were shaken for 20 min (JANKE & KUNKEL KS 501 D IKA, Staufen, Germany). Subsequently, the extraction kit (Cat# 5982-5650CH, Agilent technologies) was added and shaken for 3 min. Then samples were centrifuged at 8500 rpm for 3 min in an ultracentrifuge (Sigma 2–16K. Sigma Laborzentrifugen GmbH, Osterode am Harz, Germany), and 6 mL of the supernatant containing the extracted mycotoxins was recovered. In a second phase, the supernatant was subjected to a purification stage, or a clean-up, with DLLME (Cat# 5982-5056CH, Agilent Technologies). Pulsed vortexing was also applied for 1 min to break salts, and an orbital vortexing step was set for a duration of 3 min to increase the purification efficiency. The samples were then centrifuged at 8500 rpm for 3 min for the second time. Once centrifuged, 1 mL of the supernatant was recovered. The extract was directly diluted with 1 mL of the mobile phase. Finally, the mycotoxin extracts were filtered into uHPLC vials through a 0.22 μm PTFE filter.

### 4.7. Mycotoxins Quantification by uHPLC

The detection and quantification of AME, AOH, TeA, and TEN mycotoxins was conducted according to Zhou et al. [73], with some modifications. A uHPLC method was used for the quantification as it is less time- and solvent-consuming, generates less residue, and provides a higher resolution than traditional HPLC methods. In brief, 4 μL of each sample was injected into an Acquity™ uHPLC H-Class system (Waters^®^, Milford, MA, USA) using a reversed-phase column (Inertsil ODS4 2 μm 2.1 × 100 mm (GL Science B.V., Eindhoven, The Netherlands). The mobile phase had two components: solvent A, an aqueous solution at 0.1% of HCOOH *v*/*v* and 0.03% of EDTA-Na_2_ *w*/*v*, and solvent B, methanol: acetonitrile (7:3). Mycotoxins were eluted using a gradient from 40% solvent B (0–5.0 min), 40–55% B (5.0–6.0 min), 55% B (6.0–8.5 min), 55–65% B (8.5–10.5 min), 65% B (10.5–12.5 min), 65–95% B (12.5–13.0 min), 95% B (13.0–15.0 min), and 95–40% B (15.0–15.5 min) to initial conditions in order to re-equilibrate the column prior to the next injection. The flow of the mobile phase was 0.3 mL min^−1^, with a working pressure of around 5000 psi, and the column was maintained at 40 °C. When not being used, the column was maintained overnight with 100% acetonitrile at 0.02 mL min^−1^. TeA and TEN were detected with a photodiode (Acquity PDA uHPLC; Waters, Milford, MA, USA), and the peak areas were integrated at 280 nm. AOH and AME were detected by fluorescence (Acquity FLR uHPLC; Waters, Milford, MA, USA), setting an excitation wavelength of 295 nm and an emission wavelength of 464 nm. Mycotoxins were identified and quantified by spectral characteristics and retention time (RT) using known concentrations of standards, as described by [73]. Under our experimental conditions, TeA and TEN were detected and integrated at 280 nm, and the RTs (in min) were: TeA (5.30) and TEN (9.27). For AOH and AME, under the fluorimetric conditions described above, the RTs (in min) were: AOH (9.12) and AME (13.41). Under the chromatographic characteristics described above, the conversion factors (pmol per injection area unit^−1^) were: TeA (1.06 × 10^−4^), TEN (1.15 × 10^−4^), AOH (7.70 × 10^−6^), and AME (1.79 × 10^−5^).

### 4.8. Statistical Analysis

The results are reported as the mean ± standard error (SE) of four independent replicates. Three-way analysis of variance (ANOVA) was used to evaluate the main effects of each variable (lettuce color, inoculation, and environmental conditions). Tukey’s post hoc test was used to find significant differences among treatments. *p*-values ≤ 0.05 were considered statistically significant. Prior to the analyses, we tested whether the assumptions made by the ANOVA test, the homogeneity of variances by Levene’s test, and normally distributed residuals by the Kolmogorov–Smirnov test were achieved. Data analyses were performed using the SPSS 22.0 software package (IBM Corp, Armonk, NY, USA). 

## 5. Conclusions

This study demonstrates that phenols act as deterrents for *A. alternata*, since the red cultivar (with higher total phenols) showed lower invasion of *A. alternata* than the green one. Additionally, the results also indicate that both lettuce cultivars upregulate SOD activity in order to defend themselves from the fungi, but the activation of the antioxidative response is insufficient to provide protection against *A. alternata*. Moreover, future climate change environmental conditions will increase fungal invasion and mycotoxin production, likely because under elevated temperature and CO_2_ conditions, N levels in plant decrease and extra carbohydrates are not directed to the phenolics pathway.

Moreover, no correlation between fungal invasion and toxin production was observed in either fresh or cold stored lettuces. Fresh green and red lettuces were able to maintain biomass production and freshness without generating toxins. However, storing lettuces at 4 °C for four days, a potential procedure expected to avoid fungal growth and toxin production, did trigger mycotoxin production, suggesting that this postharvest strategy needs to be controlled. In summary, the results of the present work point to phenolic content as a key trait to select when seeking to control for this fungus infection and its mycotoxin production. However, further investigations are needed to find appropriate postharvest practices that could help to reduce mycotoxin contamination in order to safeguard animal and human health in expectation of future climatic conditions.

## Figures and Tables

**Figure 1 plants-12-01304-f001:**
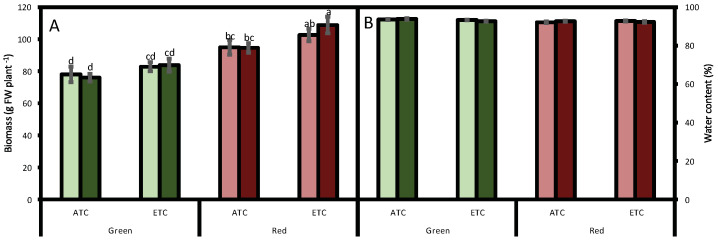
Effect of *A. alternata* inoculum and environmental conditions on the fresh biomass (**A**) and water content (**B**) of two differently-colored fresh lettuces. Light green bars represent green fresh lettuces grown without inoculum (Aa−), dark green bars represent green fresh lettuces grown with inoculum (Aa+), light red bars represent red fresh lettuces grown without inoculum (Aa−), and dark red bars represent red fresh lettuces grown with inoculum (Aa+). (ATC) refers to lettuces grown under the present climatic scenario and (ETC) to those grown under a climate change scenario. Each value represents the mean ± standard error (n = 8). Significant differences (*p* < 0.05) are indicated by different letters.

**Figure 2 plants-12-01304-f002:**
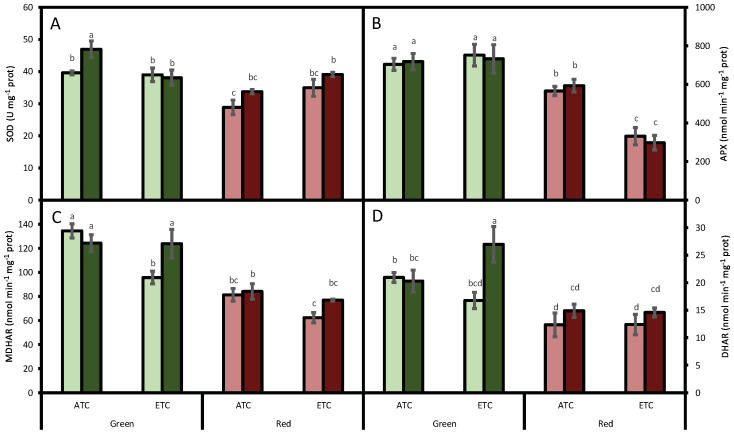
Effect of *A. alternata* inoculum and environmental conditions on the superoxide dismutase (SOD, (**A**)), ascorbate peroxidase (APX, (**B**)), monodehydroascorbate reductase (MDHAR, (**C**)), and dehydroascorbate reductase (DHAR, (**D**)) activities of two differently-colored fresh lettuces. Light green bars represent green fresh lettuces grown without inoculum (Aa−), dark green bars represent green fresh lettuces grown with inoculum (Aa+), light red bars represent red fresh lettuces grown without inoculum (Aa−), and dark red bars represent red fresh lettuces grown with inoculum (Aa+). (ATC) refers to lettuces grown under the present climatic scenario and (ETC) to those grown under a climate change scenario. Each value represents mean ± standard error (n = 4). Significant differences (*p* < 0.05) are indicated by different letters.

**Figure 3 plants-12-01304-f003:**
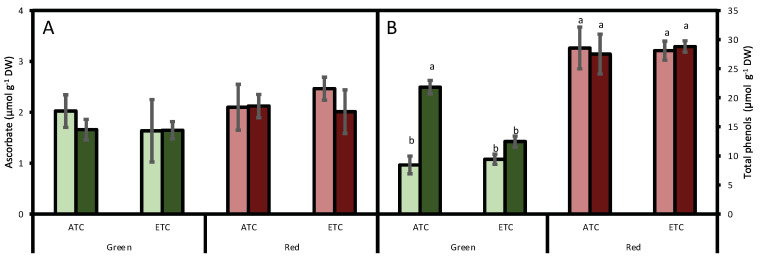
Effect of *A. alternata* inoculum and environmental conditions on ascorbate (**A**) and total phenols (**B**) of two differently-colored fresh lettuces. Light green bars represent green fresh lettuces grown without inoculum (Aa−), dark green bars represent green fresh lettuces grown with inoculum (Aa+), light red bars represent red fresh lettuces grown without inoculum (Aa−), and dark red bars represent red fresh lettuces grown with inoculum (Aa+). (ATC) refers to lettuces grown under the present climatic scenario and (ETC) to those grown under a climate change scenario. Each value represents mean ± standard error (n = 4). Significant differences (*p* < 0.05) are indicated by different letters.

**Table 1 plants-12-01304-t001:** *Alternaria alternata* genome copies and mycotoxin detection in inoculated lettuces.

	Fresh		Cold Stored	
	Green	Red	Green	Red
ATC	108	6	29	2
	21	18	48	5
	1	7	2452	148
	95	35	122 ^a^	26
	*56.5 ± 26.5*	*16.4 ± 6.8*	*662.8 ± 596.7*	*45.1 ± 34.6*
ETC	108	1	4	1
	115	0	16 ^b^	79
	2452	0	37 ^c^	3
	12	2	14	0
	*671.7 ± 565.2*	*0.7 ± 0.5*	*17.7 ± 5.4*	*20.8 ± 18.3*

The qPCR results for the fungal abundance (genome copies) in four replicated samples per condition are shown, and the mean values and standard errors are indicated in italics. The studied conditions include: present climatic scenario (ATC) and climate change scenario (ETC); green cultivar (Batavia) and red cultivar (oak leaf); and different postharvest handling, including fresh lettuce (samples analyzed immediately after harvest) and cold stored lettuce (samples maintained at 4 °C for four days after harvesting). ^a, b, c^ letters refer to mycotoxin concentration in the lettuces. ^a^ TeA (0.06 mg kg^−1^), TEN (0.12 mg kg^−1^); ^b^ TeA (0.40 mg kg^−1^), TEN (0.54 mg kg^−1^); ^c^ TeA (0.32 mg kg^−1^), TEN (0.27 mg kg^−1^).

## Data Availability

Data will be made available upon request.

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
