# Peer review of "Stress Response to Climate Change and Postharvest Handling in Two Differently Pigmented Lettuce Genotypes: Impact on Alternaria alternata Invasion and Mycotoxin Production"

_plants, 2023, doi:10.3390/plants12061304_

Round 1

Reviewer 1 Report

Manuscript "Stress response to climate change and postharvest handling in two differently pigmented lettuce genotypes: impact on Alternaria alternata invasion and mycotoxin production" is very interesting.

General comments:
Authors explored lettuce defense mechanisms to find traits that can make cultivars more resistant to A. alternata infection.
Authors addressed how future climate change scenario would affect the growth of this fungus and its toxin production, and if it would impact differently depending on the host cultivar.
Authors studied the potentiality of low-temperature storage as a postharvest handling strategy to reduce A. alternata growth and toxin production.
These are all very interesting and important issues.

Detailed comments:
The authors conducted a three-way ANOVA. Unfortunately, the paper lacks a description and discussion of the dual and triple interaction.

My suggestion:
Lines 162-163: "-1" up index.
L166: "p > 0.05". Please provide the exact value of p.
L186: Please provide the exact value of p.
L197: Please provide the exact value of p.
L202: Please provide the exact value of p.
L203: Please provide the exact value of p.
L204: Please provide the exact value of p.
L391: "A. alternata" - italic.

Paper needs minor revision.

Author Response

Manuscript "Stress response to climate change and postharvest handling in two differently pigmented lettuce genotypes: impact on Alternaria alternata invasion and mycotoxin production" is very interesting.

General comments:
Authors explored lettuce defense mechanisms to find traits that can make cultivars more resistant to A. alternata infection.
Authors addressed how future climate change scenario would affect the growth of this fungus and its toxin production, and if it would impact differently depending on the host cultivar.
Authors studied the potentiality of low-temperature storage as a postharvest handling strategy to reduce A. alternata growth and toxin production.
These are all very interesting and important issues.

Detailed comments:
(RevS1) The authors conducted a three-way ANOVA. Unfortunately, the paper lacks a description and discussion of the dual and triple interaction.
(AutR1) Yes, the reviewer is right. In most of the cases, the interactions were not significant, that is the reason why we did not add those interactions. But taking into account the suggestions made by the reviewer, we have added the interactions that were significant for each parameter in the results section. The interactions between environmental conditions and color detected for SOD and APX are not discussed in the Discussion section, since under ETC conditions, in our opinion, they do not add relevant extra information.

My suggestion:
(RevS2) Lines 162-163: "-1" up index.
(AutR2) We agree and we have changed it (P4, lines 172-173).

(RevS3) L166: "p > 0.05". Please provide the exact value of p.
(AutR3) We agree and we have changed it (P4, line 176).

(RevS4) L186: Please provide the exact value of p.
(AutR4) We agree and we have changed it (P5, line 199).

(RevS5) L197: Please provide the exact value of p.
(AutR5) We agree and we have changed it (P5, line 212).

(RevS6) L202: Please provide the exact value of p.
(AutR6) We agree and we have changed it (P6, line 218).

(RevS7) L203: Please provide the exact value of p.
(AutR7) We agree and we have changed it (P6, line 219).

(RevS8) L204: Please provide the exact value of p.
(AutR8) We agree and we have changed it (P6, line 219).

(RevS9) L391: "A. alternata" - italic.
(AutR9) We agree and we have changed it (P10, line 413).

Paper needs minor revision.

Reviewer 2 Report

The manuscript titled” Stress response to climate change and postharvest handling in 2 two differently pigmented lettuce genotypes: impact on Alternaria alternata invasion and mycotoxin production” is good and the authors paid time and efforts to introduce their results well, but it still need to check for grammar errors and spell even more you must reply on questions and comments listed below before the manuscript goes further.

-        Line 28 the red cultivar. Please add the plant name???

-        Lines 31-35 : could be changed to “Finally, while the abundance of the fungi was maintained at similar levels after keeping the lettuces for four days at 4 ºC, this postharvest handling triggered TeA and TEN mycotoxin synthesis, but only in the green cultivar. Therefore, results demonstrated that invasion and mycotoxin production are cultivar- and temperature-dependent.”

-        Line 54 : “to be susceptible of infection by Alternaria” change to “to be susceptible to infection by Alternaria”

-        Line 376: Seedlings were covered with plastic bags for 48 h to maintain suitable humidity (65–70%)> I wonder how they covered, and the authors planted in controlled growth room????

-        The introduction section is too long could be shortened.

-        Line 378 : lettuces . I think it could be lettuce plants or what you mean?

-        Line 382 : the word “scenario” the authors write it several times in the text. I think it not scientific to said. It could conclude in a film or story not in plant pathology.

-        The authors talked about the A. alternata abundance measurements in the lettuce plants were reserved in liquid nitrogen. Is that ordinary to calculate the abundance after preservation.??

-        Line 391 : “air dried barley grains showing early blight symptom” is that sentence correct the Alternaria alternata did not cause early blight in barley. How it cause even why the authors did not isolate the pathogen from the lettuce if the Alternaria made problems in lettuce as you said.

-        Line 394: how the authors morphologicaly identify the Alternaria genus. where is the manual reference.

-        Line 406: the DNA sequences not DNA only

-        Line 401: A. alternata specific primers AAF2 (5′-TGCAATCAGCGTCAGTAACAAAT-3′) and AAR3 (5′- 402 ATGGATGCTAGACCTTTGCTGAT-3′) these primers I used before, and they were not specific and amplify also A. solani species at the same length.

-        Line 407: is it one isolate or more. It confused.

-        In results: Where the accession number of the isolate you used and where is the referred paragraph in that section.

-        Line 417: it is not necessary to include the enzyme IUBMB Biochemical Nomenclature

-        Line 444: what you mean” thanks………. Is that correct?

-        Line 461: according to “ write the reference name ‘

-        Discussion section” remove the titles embedded in this section.”

-        Results : Table 1. Remove “The qPCR results for the fungal abundance (genome copies) in four replicated samples per condition are shown, and the mean values and standard errors are indicated in italics. The studied conditions include Present climatic scenario (ATC) and climate change scenario (ETC); Green cultivar (Batavia) and Red cultivar (Oak leaf); and different postharvest handling, including fresh lettuce (samples 159 analyzed right after harvest) and cold stored lettuce (samples maintained at 4 ºC during four days 160 after harvesting). a , b , c letters refer to mycotoxin concentration in lettuces” and put it under the table

-        Why the authors did not make HPLC analysis for the lettuce leaves instead of the enzyme proceedings or even quantify the total phenolic compounds .

Author Response

The manuscript titled” Stress response to climate change and postharvest handling in 2 two differently pigmented lettuce genotypes: impact on Alternaria alternata invasion and mycotoxin production” is good and the authors paid time and efforts to introduce their results well, but it still need to check for grammar errors and spell even more you must reply on questions and comments listed below before the manuscript goes further.

(RevS10) Line 28 the red cultivar. Please add the plant name???

(AutR10) We agree and we have added that information (P1, lines 28-29).

(RevS11) Lines 31-35: could be changed to “Finally, while the abundance of the fungi was maintained at similar levels after keeping the lettuces for four days at 4 ºC, this postharvest handling triggered TeA and TEN mycotoxin synthesis, but only in the green cultivar. Therefore, results demonstrated that invasion and mycotoxin production are cultivar- and temperature-dependent.”

(AutR11) We agree and we have rewritten it (P1, lines 32-36).

(RevS12) Line 54 : “to be susceptible of infection by Alternaria” change to “to be susceptible to infection by Alternaria”

(AutR12) We agree and we have changed it (P2, line 56).

(RevS13) Line 376: Seedlings were covered with plastic bags for 48 h to maintain suitable humidity (65–70%)> I wonder how they covered, and the authors planted in controlled growth room????

(AutR13) We put a plastic bag to each lettuce. We performed the experiment in lettuce plants sown in pots in an environmentally controlled growth chamber to be able to put the plants under future climatic conditions. We have rewritten this part in the text in order to be clearer (P9, lines 380-381).

(RevS14) The introduction section is too long could be shortened.

(AutR14) We shortened the introduction (by 5%). We considered its length to be adequate now.

(RevS15) Line 378 : lettuces . I think it could be lettuce plants or what you mean?

(AutR15) We agree and we have changed it (P9, line 399).

(RevS16) Line 382 : the word “scenario” the authors write it several times in the text. I think it not scientific to said. It could conclude in a film or story not in plant pathology.

(AutR16) Even if scenario is used in some non-scientific field, it is also used in the scientific field related to climate change, as can be seen in the following papers; therefore, we prefer to leave it as it is:

Moss, R., Edmonds, J., Hibbard, K. et al. The next generation of scenarios for climate change research and assessment. Nature 463, 747–756 (2010). https://doi.org/10.1038/nature08823

Guivarch, C., Le Gallic, T., Bauer, N. et al. Using large ensembles of climate change mitigation scenarios for robust insights. Nat. Clim. Chang. 12, 428–435 (2022). https://doi.org/10.1038/s41558-022-01349-x

Prieto-Benítez, S.; Ruiz-Checa, R.; González-Fernández, I. et al. Ozone and Temperature May Hinder Adaptive Capacity of Mediterranean Perennial Grasses to Future Global Change Scenarios. Plants 2023, 12, 664. https://doi.org/10.3390/plants12030664

Dorado, F.J.; Alías, J.C.; Chaves, N.; Solla, A. Warming Scenarios and Phytophthora cinnamomi Infection in Chestnut (Castanea sativa Mill.). Plants 2023, 12, 556. https://doi.org/10.3390/plants12030556

Pollastri, S.; Velikova, V.; Castaldini, M. et al. Isoprene-Emitting Tobacco Plants Are Less Affected by Moderate Water Deficit under Future Climate Change Scenario and Show Adjustments of Stress-Related Proteins in Actual Climate. Plants 2023, 12, 333. https://doi.org/10.3390/plants12020333

(RevS17) The authors talked about the A. alternata abundance measurements in the lettuce plants were reserved in liquid nitrogen. Is that ordinary to calculate the abundance after preservation.??

(AutR17) Freezing in liquid N at -196 °C and storage at -80 °C is a standard procedure for preserving samples, including DNA, until analysis.

Leiminger, J., Bäßler, E., Knappe, C. et al. (2015). Quantification of disease progression of Alternaria spp. on potato using real-time PCR. Eur. J. Plant Pathol. 141: 295–309. doi: 10.1007/s10658-014-0542-2

Abdullah AS, Turo C, Moffat CS, et al. (2018). Real-Time PCR for Diagnosing and Quantifying Co-infection by Two Globally Distributed Fungal Pathogens of Wheat. Front. Plant Sci. 9:1086. doi: 10.3389/fpls.2018.01086

Chen X, Chen X, Tan Q, et al. (2022). Selection of potential reference genes for RT-qPCR in the plant pathogenic fungus Colletotrichum fructicola. Front. Microbiol. 13:982748. doi: 10.3389/fmicb.2022.982748

(RevS18) Line 391 : “air dried barley grains showing early blight symptom” is that sentence correct the Alternaria alternata did not cause early blight in barley. How it cause even why the authors did not isolate the pathogen from the lettuce if the Alternaria made problems in lettuce as you said.

(AutR18) It was an error from our part. The Alternaria isolate was extracted from barley showing premature ripening of the crop and darkening of the ears at harvest, a disease usually attributed to Alternaria spp and Cladosporium spp (Kelly et al. 2020). We have modified the text accordingly (P10, lines 413-414).

Kelly C, Bill C, Bryson R, Jellis G, Tonguc L. 2020 The encyclopedia of cereal diseases. HGCA & BASF.

We could not find a commercial strain of Alternaria alternata isolated from lettuce. We searched on the Spanish Type Culture Collection, where we only found one strain isolate from persimmon and this species is not close to lettuce. We also searched on Westerdijk Fungal Biodiversity Institute, where its curator Dr. Gerard Verkley could not find a suitable candidate between the 56 Alternaria entries within their collection. Therefore, first we conducted an experiment with the barley isolate to test whether it could infect lettuce plants. Since we detected symptoms, we decided to work with this isolate. Alternaria species and their respective hosts are currently under investigation. The function and biosynthesis of Alternaria alternata host-specific toxins (HST) has gained much attention and considerable advances have been made recently. The general strategies employed by the Alternaria alternata pathogenic strains to attack their hosts are similar, however, the responses of the various hosts against these attacks are very diverse. However, the HST are linked to pathogenicity but they are not required for fungal normal growth but have an adaptive advantage in some habitats to the individual specie. Alternaria isolates with the same HST profile can have a range of hosts belonging to different plant crop families (Meena et al 2017).

Considering the pressure established by genetic improvement for varietal selection, it is possible that the same fungal isolate may carry differences in varietal pathogenicity and virulence but not in mycotoxin development dependent on climatic conditions. Therefore, we consider that the host, barley or lettuce, would not affect the results found in this work.

Meena M.h, Gupta Sanjay K., Swapnil P., Zehra A., Dubey M.K., Upadhyay R.S. 2017. Alternaria Toxins: Potential Virulence Factors and Genes Related to Pathogenesis. Frontiers in Microbiology 8 1451- doi 10.3389/fmicb.2017.01451   

(RevS19) Line 394: how the authors morphologicaly identify the Alternaria genus. where is the manual reference.

(AutR19) The isolates were examined under a stereomicroscope taking into account that the key taxonomic feature of the genus Alternaria is the production of large, multicellular, dark-colored (melanized) conidia with longitudinal as well as transverse septa (phaeodictyospores) (Thomma, 2003). We have added that information in the text (P10, lines 417-419).

Thomma, B.P., 2003. Alternaria spp.: from general saprophyte to specific parasite. Mol. Plant Pathol. 4, 225-236.

(RevS20) Line 406: the DNA sequences not DNA only

(AutR20) We agree and we have changed it (P10, line 432).

(RevS21) Line 401: A. alternata specific primers AAF2 (5′-TGCAATCAGCGTCAGTAACAAAT-3′) and AAR3 (5′- 402 ATGGATGCTAGACCTTTGCTGAT-3′) these primers I used before, and they were not specific and amplify also A. solani species at the same length.

(AutR21) This terminology has been used in other published manuscripts for this primers (Konstantinova et al 2002, doi:10.1017/s0953756201005160; Meena et al 2016, doi: 10.3389/fpls.2016.01408; and Basim et al 2017, doi:10.1016/j.cropro.2016.10.013.) and that is why we used it, but we agree with the reviewer that the primers don’t seem to be as specific (and have removed that terminology from the paper, p10 line 427), as in the blast search some hits matched with other Alternaria species (e.g. tenuissima, solani… see below) with same e value and percentage of identity. However, the sequence primarily matched to A. alternaria (according to megablast and blastn results), the morphological characterization suggested the isolate to be A. alternata, and importantly, AltAlt dtec-qPCR Test kit that uses primers designed to specifically amplify A. alternata, excluding other species from the genera (solani among them), efficiently amplified A. alternata in the samples inoculated with the isolate.

Megablast result for the isolate sequence:

(RevS22) Line 407: is it one isolate or more. It confused.

(AutR22) It refers to one isolate, the text has been modified (P10, line 433-435).

(RevS23) In results: Where the accession number of the isolate you used and where is the referred paragraph in that section.

(AutR23) The isolate sequence has been uploaded to NCBI, accession number OQ434290 and this information has been included in material and methods (P10, line 433-435). We selected to make this data public upon manuscript publication.

(RevS24) Line 417: it is not necessary to include the enzyme IUBMB Biochemical Nomenclature

(AutR24) We agree and we have changed it (P10, lines 445-447).

(RevS25) Line 444: what you mean” thanks………. Is that correct?

(AutR25) We agree and we have changed it (P11, line 472).

(RevS26) Line 461: according to “write the reference name ‘

(AutR26) We agree and we have changed it (P11, line 489).

(RevS27) Discussion section” remove the titles embedded in this section.”

(AutR27) We consider that the discussion is easier to follow with the subdivisions, since the manuscript has two parts, one that shows the results and discussion obtained for fresh lettuce samples (part one), and another part that shows the information related to cold stored lettuces (second part).

(RevS28) Results: Table 1. Remove “The qPCR results for the fungal abundance (genome copies) in four replicated samples per condition are shown, and the mean values and standard errors are indicated in italics. The studied conditions include Present climatic scenario (ATC) and climate change scenario (ETC); Green cultivar (Batavia) and Red cultivar (Oak leaf); and different postharvest handling, including fresh lettuce (samples 159 analyzed right after harvest) and cold stored lettuce (samples maintained at 4 ºC during four days 160 after harvesting). a , b , c letters refer to mycotoxin concentration in lettuces” and put it under the table

(AutR28) We agree and we have changed it (P4, line 166-171).

(RevS29) Why the authors did not make HPLC analysis for the lettuce leaves instead of the enzyme proceedings or even quantify the total phenolic compounds .

(AutR29) We wanted to verify our assumption that the red lettuce contained higher phenolic content and also, we wanted to test whether the plants responded to inoculation increasing the synthesis of these compounds. Therefore, we believe that the measurement we conducted was the appropriate and simpler way, while conducting a phenolic profile would have given much more information but was beyond the scope of our goals.

Reviewer 3 Report

My comments can be found in the attached MS.

Author Response

Unless stated below, we addressed the changes proposed in the pdf, please see the manuscript revised for all those changes.

(RevS30) Line 33: Please insert space here and in the rest of the document.

(AutR30) According to the SI it is set correctly as 4 °C. https://www.nist.gov/pml/owm/si-units-temperature

(RevS31) Line 139: spores?

(AutR31) In the method to determine the presence and abundance of A. alternata we analyzed the DNA lysed from fungal cells, but we can´t distinguish between spores, mycelia or other fungal fragments; therefore, it would not be adequate to include “spores” in here.

(RevS32) Line 226: fungal spores?

(AutR32) As mentioned for the previous comment, we think that it is better to let as written.

(RevS33) Line 300: Please be specific and mention the type of sugar

(AutR33) In the references, they used the sugar term without specifying it. Consequently, we cannot assure the specific sugar they are referring to, but we believe it is sucrose or glucose, since they are the most abundant sugars in leaf tissues.

Horsfall, James G., and A. E. Dimond. “Interactions of Tissue Sugar, Growth Substances, and Disease Susceptibility.” Zeitschrift Für Pflanzenkrankheiten (Pflanzenpathologie) Und Pflanzenschutz, vol. 64, no. 7/10, 1957, pp. 415–21. http://www.jstor.org/stable/43231735.

Abuley, I.K.; Nielsen, B.J.; Hansen, H.H. The influence of timing the application of nitrogen fertilizer on early blight (Alternaria solani). Pest Manage. Sci. 2019, 75, 1150-1158, doi:10.1002/ps.5236.

(RevS34) Line 391: So the pathogen is a barley isolate, why it was not isolated from lettuce? If it were a lettuce isolate, it would have affected the results differently.

(AutR34) We could not find a commercial strain of Alternaria alternata isolated from lettuce. We searched on the Spanish Type Culture Collection, where we only found one strain isolate from persimmon and this species is not close to lettuce. We also searched on Westerdijk Fungal Biodiversity Institute, where its curator Dr. Gerard Verkley could not find a suitable candidate between the 56 Alternaria entries within their collection. Therefore, first we conducted an experiment with the barley isolate to test whether it could infect lettuce plants. Since we detected symptoms, we decided to work with this isolate. Alternaria species and their respective hosts are currently under investigation. The function and biosynthesis of Alternaria alternata host-specific toxins (HST) has gained much attention and considerable advances have been made recently. The general strategies employed by the Alternaria alternata pathogenic strains to attack their hosts are similar, however, the responses of the various hosts against these attacks are very diverse. However, the HST are linked to pathogenicity but they are not required for fungal normal growth but have an adaptive advantage in some habitats to the individual specie. Alternaria isolates with the same HST profile can have a range of hosts belonging to different plant crop families (Meena et al 2017).

Considering the pressure established by genetic improvement for varietal selection, it is possible that the same fungal isolate may carry differences in varietal pathogenicity and virulence but not in mycotoxin development dependent on climatic conditions. Therefore, we consider that the host, barley or lettuce, would not affect the results found in this work.

Meena M.h, Gupta Sanjay K., Swapnil P., Zehra A., Dubey M.K., Upadhyay R.S. 2017. Alternaria Toxins: Potential Virulence Factors and Genes Related to Pathogenesis. Frontiers in Microbiology 8 1451- doi 10.3389/fmicb.2017.01451   

(RevS35) Line 397: Fungal instead of Fungi

(AutR35) "NZY Plant/Fungi gDNA Isolation kit" is the brand name, therefore, we think that it is better to let as written.

Round 2

Reviewer 2 Report

Based on the authors' response, it seems that the manuscript has been improved and is ready for acceptance in its current form.

Author Response

We thank the reviewer for his/her input in improving the manuscript

Reviewer 3 Report

Some minor changes need to be made and they can be found in the attached MS.

Author Response

Except of the comments that is exposed below, we have addressed the rest of the comments in the text, please see the manuscript revised for all those changes.

It is one comment that also appeared in the first review.

(RevS in the first review) Line 33: Please insert space here and in the rest of the document.

(AutR in the first review) According to the SI it is set correctly as 4 °C. https://www.nist.gov/pml/owm/si-units-temperature

(RevS in this new review) Line 34: I see that you addressed this, but there is an extra space between the number and °; the ° and C are like conjoined twins; I hope you got my point.

(AutR) We might have misunderstood the previous request... Sorry, we are afraid we are confused now with what is what the reviewer meant. After checking other published manuscripts in Plants journal and taking into account the general rule of the International Bureau of Weights and Measures (BIPM) what states that the numerical value always precedes the unit, and a space is always used to separate the unit from the number, we believe that the way it is written currently in the manuscript is the right one (4 °C). If we are not right, could reviewers please specify which is the correct one? (4 °C, 4° C, 4 ° C)